# Little Brands, Big Profits? Effect of Agricultural Geographical Indicators on County-Level Economic Development in China

**Zhuang Zhang [1], Qiuxia Yan [2], Hao Zheng [3], Mengqing Zeng [4] and Youhua Chen [4,5,\*]**

1   School of Business Administration, Zhongnan University of Economics and Law, Wuhan 430073, China; zhangzhuang@stu.zuel.edu.cn
2   School of Business Administration, South China University of Technology, Guangzhou 510642, China; bmyanqx@mail.scut.edu.cn
3   School of Statistics, Institute of Quantitative Economics, Huaqiao University, Xiamen 361021, China; zhenghao9720@163.com
4   College of Economics and Management, South China Agricultural University, Guangzhou 510642, China; mqzeng413@stu.scau.edu.cn
5   Research Center for Green Development of Agriculture, South China Agricultural University, Guangzhou 510642, China
\*   Correspondence: chenyhua214@163.com

**Abstract:** AGIs (agricultural geographical indicators) are effective quality signals that can improve market welfare, but few studies have investigated the impact of AGIs on economic development. To fill this gap, this paper explores the impact of AGIs on per capita GDP and its mechanisms, according to country-level data in China from 2000 to 2018. For every additional AGI in the country, GDP per capita increased by 0.2–0.4%. Our conclusion remained reliable after various robustness tests. These effects were more salient in western areas, the main grain-producing areas, and settled areas. AGIs related to aquatic environments, animal husbandry, and planting products promoted economic development most significantly. For these effects, encouraging an increase in agricultural value (improving the quantity and quality of products) and promoting the agglomeration of populations, capital, and enterprises in the agricultural sector were the main mechanisms.

**Keywords:** agricultural geographical indicators; economic development; agricultural value-added; agglomeration effect

## 1. Introduction

Geographical indications are special intellectual property rights based on the long-term practices and uniqueness in a certain region. Europe first used AGIs (agricultural geographical indicators) to protect and differentiate local agricultural products [1]. Since 1995, the World Trade Organization (WTO) has formally recognized AGIs as intellectual property and AGIs have received widespread attention worldwide [2]. The WTO defines AGIs as goods produced in specific regions, locations, or territories of member countries. Commodities with AGIs typically have specific qualities, reputations, or other characteristics, largely attributed to their geographical origins [3,4]. Considering the effects of AGIs on economic development, Europe, the United States, and Australia have established leading positions in AGI certification in recent years. However, AGI certification is relative backward in Asian countries. Faced with this situation, in 2008, the Chinese Ministry of Agriculture prioritized AGIs as a key means of promoting regional economic development. In China, how do AGIs affect aggregated economic development? What are the corresponding mechanisms? Few studies have provided rich knowledge related to these questions, which encouraged this article.

Previous studies have mostly analyzed how AGIs affected agricultural products from the standpoints of international trade, market welfare, and legal conflicts [4–6]. However,

few studies have discussed the effects of AGIs on regional economic development from a macro-perspective.

Firstly, AGIs have an impact on legal conflicts. AGIs are a means of protecting intellectual property rights, and are an important tool for encouraging the development of agricultural products. The recognition of GIs for agricultural products must meet the following conditions: the goods and products must come from a specific area, and the reputation, quality, and characteristics of the goods should mainly be attributed to their geographical environmental factors [4]. However, AGI certification in cultural and service products in rural areas has often has some difficulties. On the one hand, the geographical attribution of some cultural services often has a wide range, and their specific attribution is controversial. On the other hand, the formation of cultural services often cannot be directly linked to geographical factors and requires strong historical evidence to prove this. This makes the recognition of GIs (geographical indicators) cumbersome. Moreover, although the WTO has provided certain explanations for the recognition of GIs, there are subtle differences in the recognition standards of GIs in different countries, which may lead to international trade conflicts.

Secondly, AGIs have an impact on market welfare. As a representation of brand and quality, AGIs have a tremendous impact on both market customers and manufacturers. Moschini's theoretical model implies that GI recognition has an impact on both producers and consumers, with consumers benefiting the most [5]. However, if AGIs cause a spike in equilibrium prices, they greatly enhance producers' marginal profits [6]. Consumers have a stronger sense of trust and readiness to buy agricultural products with AGIs because of their distinct flavor, color, texture, and other qualities [7], and they are willing to pay higher prices for them [8]. GIs for wine in Spain, France, and Italy encourage customers to pay higher prices, while simultaneously reducing adverse competition for growers [9]. AGIs for coffee in Kenya and Colombia enable producers to increase their revenue [10]. Of course, some research indicates that GIs have little impact on farmers.

Thirdly, AGIs have an impact on trade. Agricultural products with AGIs often undergo strict scrutiny and are generally considered high-quality products [11]. Because of this attribute, GIs are often seen as tools to promote trade [12,13]. By analyzing bilateral wine export data from France, Italy, and Spain during the period 2010–2013 [14], it was found that AGIs were beneficial for price premiums of products and promoted wine trade [15] and that AGIs in France could help products penetrate overseas markets. Additionally, mutual recognition of GIs between countries can effectively improve the export quality of agricultural products [16]. Of course, the impact of AGIs on international trade is also constrained by consumer preferences for products [17].

In summary, previous studies have mostly discussed the legal recognition of AGIs or their impact on market welfare, with few works concerned with their economic contribution. Therefore, we collected the data on agricultural products with AGIs in China from 2000 to 2018, and matched them with country-level statistical data, to investigate the impact and mechanism of GIs of agricultural products on country-level economic development.

The rest of the article is organized as follows: the second part is the policy background and theoretical hypotheses; the third part is the research design; the fourth part is the results; and the final part is the conclusions, discussions, policy implications, and limitations.

## 2. Policy Background and Theoretical Hypotheses

### 2.1. Policy Background

In addition to the Trademark Law, in 2000, the Agricultural Law of China formally introduced the criteria for identifying GIs for agricultural products. Subsequently, China promulgated the "Regulations on the Protection of Geographical Indication Products", specifically aimed at regulating and protecting intellectual property rights related to agricultural products. But, differently from other commodities, there are specific regulations governing the application, registration, and supervision of the use of GIs for agricultural products, and the supervision of the two departments of industry and commerce and

quality inspection is often not entirely effective. Therefore, depending on the advantages of familiarity and convenience in the process of agricultural management, such as the special qualities of specific products in certain geographical areas, delineation of the geographical scope of products, and monitoring of the production process, the department believes that it is relatively efficient and less controversial to carry out the certification and management of GIs for agricultural products.

In 2007, to systematically regulate the registration and use of GIs for agricultural products, ensure the quality and characteristics of geographical indication agricultural products, and enhance the market competitiveness of agricultural products, the former Ministry of Agriculture issued the Management Measures for AGIs of Agricultural Products, which stipulated the requirements and procedures for the registration of GIs of agricultural products. Subsequently, the Ministry of Agriculture began to accept applications, and issue several specialized AGIs for agricultural products. According to this method, the applicant for the registration of AGIs of agricultural products is a farmer's professional cooperative economic organization, industry association, or other organizations selected by the local people's government at or above the county level and based on specific conditions. In July 2008, the first batch of 28 geographical indication products for agricultural products was promulgated. On 8 August 2008, Announcement No. 1071 of the Ministry of Agriculture of the People's Republic of China was issued, which stipulated the registration procedures and usage standards for GIs of agricultural products. From 2008 to 2020, the geographical indication registration system for agricultural products in China was gradually improved, and many agricultural products that met the registration system obtained geographical indication registration. As of January 2021, the Ministry of Agriculture and Rural Affairs had registered 3125 AGIs for agricultural products.

### 2.2. Theory Hypothesis

#### 2.2.1. The Direct Impact of GIs on the Economy

AGIs have a direct impact on economic value-added. AGIs, as an important market signal, can reduce the degree of information asymmetry between buyers and sellers [18]. Firstly, as an effective marketing management tool [19], AGIs can increase consumer willingness to pay. Agostino and Trivieri found that GIs for wine can make consumers more willing to pay higher prices [9]. Secondly, AGIs are beneficial in motivating farmers and enterprises to produce according to quality standards, improving product quality [20], and achieving market premiums [21]. Barjolle et al. argued that AGIs in Kenya and Colombia not only improve coffee quality, but also provide producers with additional benefits [10]. Thirdly, AGIs have obvious cultural attributes, which are conducive to driving the development of regional culture, tourism, and catering industries. For example, special food routes have become important tourism projects [22]. Based on AGIs, regions develop various specialty products and dishes in food stores, grocery stores, and restaurants [23] to attract tourists [24]. Based on this, we can propose the first hypothesis.

**H1:** *AGIs can drive regional economic growth.*

#### 2.2.2. The Mechanism of GIs Affecting Economic Growth

AGIs mainly affect regional economic growth through two paths: industrial value-added and factor agglomeration. Firstly, with respect to the industrial value-added effect of AGIs, agricultural products with AGIs are generally more likely to form regional brands [25,26], creating added value for retailers and consumers [27]. AGIs link products, places of origin, and consumption, reducing consumer-choice costs and increasing consumer willingness to pay [28]. Moreover, high-quality geographical-indication products have expanded consumer demand for products and improved the bargaining power of manufacturers [29]. Creating AGIs is conducive to establishing agricultural brands, building up product reputation, achieving brand premiums [30], promoting regional agricultural product production, processing, and exports [9,29], and driving farmers to work and im-

prove regional income levels [30–32]. The geographical symbols of AGIs are conducive to achieving an emotional premium. Geographical indications of agricultural products are often important carriers of hometown emotions [33]. During people's lives, products with hometown imprints are often prone to emotional resonance [34,35], as their daily consumption often shows loyalty to the AGIs of their home town [36,37]. Based on this, we can propose the second hypothesis.

**H2:** *AGIs drive regional economic growth through industrial value-added.*

Secondly, with respect to the economic agglomeration effect of AGIs, the two major characteristics of AGIs determine their economic agglomeration effect. One is the siphoning effect caused by the irreplaceable regional characteristics of agricultural products. Unlike other commodities, geographical indication agricultural products must be produced in specific regions [11], and various types of production and services related to AGIs are also subject to regional restrictions [38,39]. This irreplaceable characteristic determines that industrial development must be centered around geography. The recognition of AGIs will significantly drive the aggregation of funds, talents, and technology to specific regions, promote agricultural industrial clusters, and improve the efficiency of agricultural production factor allocation. Hilal et al. argue that the recognition and development of AGIs can offset the trend of urban migration in rural areas and help preserve local economic and social capital [40]. The second is the industrial correlation effect generated by AGIs. The identification of GIs alleviates the asymmetry of market information, which is beneficial for the formation of stable contractual relationships between farmers, enterprises, and enterprises. Relying on the word-of-mouth effect of the previous stage, this attracts factor agglomeration and forms a complete upstream industrial chain [41,42]. At the same time, it also relies on the advantages of upstream industries to form quality effects and achieve the convergence of downstream industrial chains [43]. Depending on the correlation between upstream and downstream industries, the identification of AGIs for agricultural products is beneficial for reducing production costs, improving production efficiency [44], and achieving economic spillovers.

**H3:** *AGIs drive regional economic growth through economic agglomeration.*

## 3. Research Design
### 3.1. Basic Model

In this paper, the per capita gross domestic product and AGIs varied in different years and counties, so we adopted a model that considers time effects and individual effects. Fixed regional characteristics (geographical location, planting habits, market distance, etc.), and nationwide policy shocks (agricultural support policies, agricultural insurance policies, and ecological compensation policies) will interfere with the impact of AGIs on per capita GDP. If nationwide policy shocks and fixed regional characteristics are not important for the effects, the random effects model can be better than the fixed effects model. Otherwise, the fixed effects model will be better. In this paper, the Hausman value was 468.71 (*p*-value < 0.01), which implies that we should employ a two-way fixed effects model to capture the economic contribution of agricultural geographical indications. The corresponding equation is as follows:

$$LnGDP_{it} = \beta_1 AGI_{it} + \Gamma'X + c_t + f_i + \varepsilon_{it} \qquad (1)$$

where $GDP_{it}$ denotes per capita gross domestic product in county *i* of year *t*, $AGI_{it}$ denotes the number of agricultural geographical indications, $\beta_1$ is corresponding estimated coefficient, *X* is the vector of control variables, $\Gamma$ is the vector of the corresponding coefficient, $c_t$ represents the year fixed effect, $f_i$ means the county fixed effect, and $\varepsilon_{it}$ is error term.

*3.2. Variables*

(1)  Explained variable

GDP per capita (gross national product per capita in a county) is used to measure economics development.

(2)  Explanatory variable

Number of geographical indications of added agricultural products in county *i* of year *t* can be used to denote AGIs.

(3)  Control variables

We controlled important variables like fiscal pressure, population, investment, and education. Fiscal pressures and competitive pressures have forced regional officials to take more aggressive measures to promote economic growth [45], although these approaches are often environmentally unfriendly [46,47]. Following Zhang et al., we used the ratio of fiscal expenditure to fiscal revenue to indicate the fiscal pressure. Investment is an important engine driving economic growth [48]. Here, we used firm fixed asset investment to characterize the firm investment intensity. The quality and quantity of human capital are the guarantee of sustainable economic development. We used the number of people receiving primary and secondary education per 100 people to represent the quality of human capital [49] and we used the number of people per unit area to represent the quantity of human capital [50]. Table 1 denotes the definition and descriptive statistics of key variables.

**Table 1.** Definition and descriptive statistics of key variables.

| Variable | N | Definition | Mean | S. D |
|---|---|---|---|---|
| GDP | 33,660 | Ln (GDP per capita in a county) | 9.487 | 0.999 |
| AGI | 33,660 | Number of geographical indications | 0.456 | 1.162 |
| Fiscal | 33,660 | Fiscal expenditure/fiscal revenue | 0.349 | 0.241 |
| Invest | 33,660 | Ln (total investment in fixed assets) | 4.551 | 2.184 |
| Density | 33,660 | Number of people per unit area | 0.033 | 0.148 |
| Educ | 33,660 | Number of people receiving primary and secondary education per 100 people | 5.467 | 1.844 |

*3.3. Data*

We mainly use the two sets of data in this paper. The first set of data is from geographical indications of agricultural products query system (http://www.anluyun.com, accessed on 13 May 2024), and provides the time, specific name, and city of agricultural geographical indications. In 2008, China recognized 282 agricultural geographical indications. By 2018, China had recognized 8716 agricultural geographical indications. The second set of data comes from the 2000–2018 county panel data, which mainly reflect the county's economy, and population-, finance-, and investment-related information.

## 4. Results

*4.1. Basic Results*

Table 2 reports the effects of AGIs on GDP per capita. Columns (1)–(6) display the effects of AGIs on GDP per capita when we added controlled variables one by one. The results indicate that ceteris paribus, the GDP per capita increased 0.44% when an AGI was is added. This demonstrates that $H_1$ could be rejected and so, AGIs were able to directly promote GDP per capita. As a symbol of good food quality, AGIs can not only effectively increase the willingness of payment for consumers [20], but also improve the revenue for manufacturers [20] and thus improve the overall market welfare [21].

**Table 2.** Effects of AGIs on GDP per capita.

|  | (1) | (2) | (3) | (4) | (5) | (6) |
|---|---|---|---|---|---|---|
| AGI | 0.0044 *** | 0.0045 *** | 0.0047 *** | 0.0051 *** | 0.0052 *** | 0.0044 *** |
|  | (0.0012) | (0.0012) | (0.0012) | (0.0012) | (0.0012) | (0.0011) |
| Fiscal |  | −0.4429 *** | −0.4400 *** | −0.4240 *** | −0.4241 *** | −0.3786 *** |
|  |  | (0.0096) | (0.0096) | (0.0095) | (0.0094) | (0.0092) |
| Educ |  |  | 0.0093 *** | 0.0119 *** | 0.0120 *** | 0.0046 *** |
|  |  |  | (0.0010) | (0.0009) | (0.0009) | (0.0009) |
| Invest |  |  |  | 0.0752 *** | 0.0770 *** | 0.0710 *** |
|  |  |  |  | (0.0024) | (0.0024) | (0.0023) |
| Density |  |  |  |  | −0.0399 *** | −0.0357 *** |
|  |  |  |  |  | (0.0084) | (0.0081) |
| Year effects | √ | √ | √ | √ | √ | √ |
| County effects | √ | √ | √ | √ | √ | √ |
| _cons | 9.4853 *** | 9.3310 *** | 9.2810 *** | 8.9300 *** | 8.9227 *** | 6.7680 *** |
|  | (0.0014) | (0.0036) | (0.0062) | (0.0126) | (0.0127) | (0.0478) |
| N | 33,660 | 33,660 | 33,660 | 33,660 | 33,660 | 33,660 |
| $R^2$ | 0.9517 | 0.9548 | 0.9550 | 0.9563 | 0.9564 | 0.9592 |

Standard errors in parentheses, *** $p < 0.01$.

Coeffecients of control variables can be demonstrated as follows: Fiscal pressure obviously inhibits GDP. Local governments often increase their fiscal revenue through various means (such as industrial adjustment, land transfer, and debt financing [51]), but the fiscal pressure has compressed the fiscal revenue of local governments and hindered the long-term economic growth. The quantity of human capital that is too dense hinders the economic development in a country, while the quality of human capital has an obvious promotion effect on economic development [52]. On the one hand, differently from big cities, county infrastructure construction is poor, and population carrying capacity is poor, so population agglomeration is not conducive to economic growth [53]. On the other hand, improving the education of residents is beneficial for innovation, intelligent upgrading of industries, and promotion of economic development [54]. Fixed-asset investment has an obvious positive effect on economic development as it promotes market integration and knowledge capital [55].

*4.2. Dynamic Effects*

Following Zhang et al. (2023) [48], the events study method was employed in this paper to carry out a parallel trend and capture the dynamic effects. The model is as follows:

$$LnGDP_{it} = \sum_{k=-10(k\neq-1)}^{10} \beta_k AGI\_dummy_{ik} + \Gamma'X + c_t + f_i + \varepsilon_{it} \tag{2}$$

where $k$ denotes the periods after or before AGI is obtained in county $i$, and $AGI\_dummy_{ik} = 1$ if the numbers are more than zero in period $k$ for the county; otherwise, $AGI\_dummy_{ik} = 0$. The nomenclatures of the other variables are consistent with Equation (1). Figure 1 capture the dynamic effects of AGIs on GDP per capita.

In Figure 1, the $Y$ axis represents the estimated coefficient, the $X$ axis represents the period point after or before AGIs are granted in a county. *before_i* ($i$ = 2, 3, . . . . . . 10) denotes the $i$ year before the policy is implemented, *after_j* ($j$ = 1, 2, . . . . . . 10) is the $j$ year after the policy is implemented, and current represents the year the policy is implemented. To avoid multicollinearity problems, the second period before policy implementation (*before_2*) is considered as the baseline group [48]. In Figure 1, the dashed line represents the 95% confidence interval, the hollow point represents the estimated coefficient corresponding to each time point, and the horizontal dashed line represents the zero value. The results in Figure 1 show that the estimated intervals of all time points before the occurrence of the policy contain 0. Therefore, before the AGIs were granted, the county economy had no obvious growth trend; that is, the assumption of parallel trend is satisfied. In the second

period, after the AGIs were obtained, the estimated interval of all time points is greater than 0 and the estimated coefficient is increasing. That is, AGIs had an obvious driving effect on economic growth in the county.

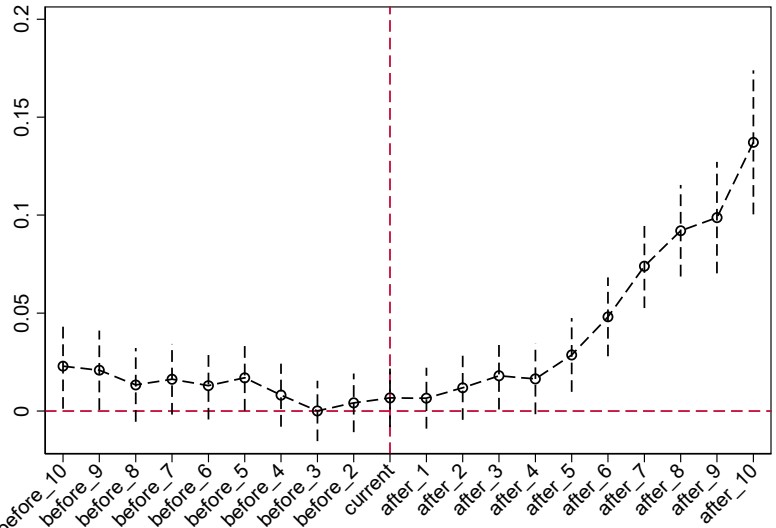

**Figure 1.** Dynamic effects.

*4.3. Robust Results*

A series of methods are used to test the robustness of our estimated coefficient. The results are outlined in Table 3.

**Table 3.** Robust results.

|  | (1) | (2) | (3) | (4) | (5) |
|---|---|---|---|---|---|
|  | **No Capital Cities** | **DID** | **PSM** | **Year > 2007** | **Lewel IV** |
| AGI | 0.0052 *** | | 0.0034 *** | 0.0026 *** | 0.0062 *** |
|  | (0.001) | | (0.001) | (0.0009) | (0.002) |
| AGI_DID | | 0.0185 *** | | | |
|  | | (0.005) | | | |
| Control variables | √ | √ | √ | √ | √ |
| Year effects | √ | √ | √ | √ | √ |
| County effects | √ | √ | √ | √ | √ |
| _cons | 6.815 *** | 6.763 *** | 6.980 *** | 7.363*** | −0.658 *** |
|  | (0.0476) | (0.0478) | (0.0931) | (0.0622) | (0.0441) |
| N | 30,718 | 33,660 | 12,582 | 19,487 | 33,660 |
| $R^2$ | 0.961 | 0.959 | 0.966 | 0.966 | 0.831 |

Standard errors in parentheses, *** $p < 0.01$.

The proportion of agricultural output value in provincial capitals was relatively low, and may have affected the distribution of samples, and thus affected the estimated coefficient. In column (1) of Table 3, we deleted this sample. This showed that the estimated coefficient was less than 0 at the significance level of 1% if we cut these samples. Therefore, extreme samples did not affect the results. In column (2) of Table 3, we define the counties who own AGIs as the experimental group. The straggle DID (difference in difference) method was used to analyze the impact of the AGIs on economic growth. We found that the economic growth rate of the regions with AGIs was 1.85% higher than other areas, and was four times the coefficient in Table 2. This does not mean that the estimated coefficients were not robust. This was because we measured the effects of AGIs, rather than added AGIs onto economic development. Obviously, the latter had less impact on economic growth.

In column (3) of Table 3, based on whether it was an experimental group or not, we tried to generate a dummy variable. Then, we used the 1:4 neighbor PSM (propensity score matching) method to cut out non-random factors and re-estimate the impact of AGIs on GDP per capita. This shows that the positive effect of AGIs on GDP per capita was still significant. In column (4) of Table 3, we just examine the impact of AGIs on GDP per capita since the AGIs policy had been carried out. The estimated result is still robust. In column (4) of Table 3, the Lewbel IV (instrumental variable) method was used to mitigate estimated bias due to missing variables. Lewbel used the heteroscedasticity of the regression equation to generate an IV, which is suitable for some tricky problems related to the IV [56]. We used this approach to correct the estimated bias. The *F* value of the first stage was 1958.830. This indicates that the weak instrumental variables problem was not obvious and the estimated coefficient is credible. The estimated coefficient in column (5) of Table 3 is consistent with the result in Table 2. It shows that missing variables do not seriously affect estimated results.

### 4.4. Heterogeneities

China has recognized AGIs nationwide; however, industrialization and the market demand is different for different agricultural products and different areas, so the impact of AGIs on economic development is also different. In Table 4, we try to dig out the effects of AGIs in different areas.

**Table 4.** Effects of AGI on GDP per capita in different areas.

|  | (1) | (2) | (3) |
|---|---|---|---|
| AGI | 0.015 *** | 0.009 *** | 0.002 * |
|  | (0.002) | (0.002) | (0.001) |
| Western area = base line |  |  |  |
| AGI × Dummy (Middle area = 1) | −0.010 *** |  |  |
|  | (0.003) |  |  |
| AGI × Dummy (Eastern area = 1) | −0.025 *** |  |  |
|  | (0.003) |  |  |
| AGI × Dummy(grains producing area = 1) |  | 0.008 *** |  |
|  |  | (0.002) |  |
| AGI × Dummy (Han nations = 1) |  |  | −0.010 *** |
|  |  |  | (0.003) |
| Control variables | √ | √ | √ |
| Year effects | √ | √ | √ |
| County effects | √ | √ | √ |
| _cons | 6.799 *** | 6.768 *** | 6.769 *** |
|  | (0.048) | (0.048) | (0.048) |
| N | 33,660 | 33,660 | 33,660 |
| $R^2$ | 0.959 | 0.959 | 0.959 |

Standard errors in parentheses, *** $p < 0.01$, * $p < 0.1$.

Table 4 shows that AGIs had a greater influence on economic development in the western region compared to the central and eastern regions. AGIs had a more noticeable effect on the economic growth of major grain-producing areas than they did on non-major grain-producing areas. AGIs had a more noticeable impact on boosting the local economy in other nations than they did on the Han nation, which accounts for 91.11% of the population in China. This can be explained as follows: The western region, the major grain-producing regions, and other-nations gathering areas have relatively low levels of marketization for agricultural production. Agricultural products are not well-known. AGIs can create a premium brand impression for consumers and more overtly support the economic growth of these regions.

Further, we analyzed the contribution of different geographical indications of agricultural products to county economy. The estimated results are shown in Figure 2.

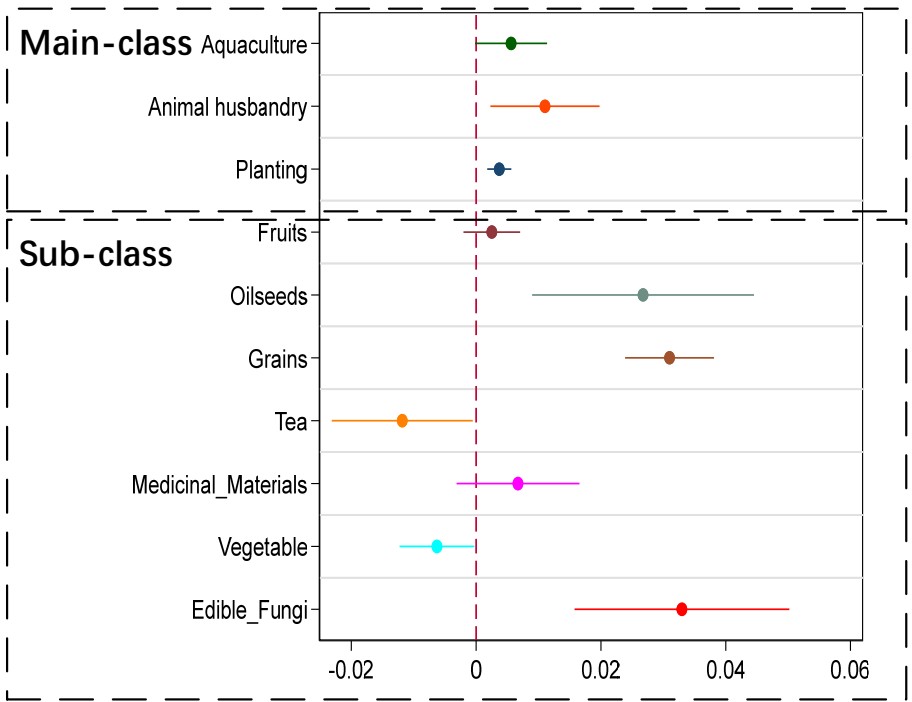

**Figure 2.** Effects of AGI on GDP per capita for different productions.

In Figure 2, the solid horizontal line shows the 95% confidence interval, and the solid point shows the impact of different AGIs on GDP per capita. It shows that main-class AGIs, related to aquatic, animal husbandry, and planting products obviously promoted economic development. Moreover, compared with planting AGIs, aquatic products and livestock AGIs boomed at a higher level. We also analyzed the impact of sub-class AGIs on the county's economy. This showed that AGIs of edible fungi, fish, meat, grain, and oil crops had more obvious effects than that those of fruit, tea, medicinal materials, vegetables, and honey. The demand curves for edible fungi, fish, meat, grain, and oil crops had strong elasticity, as these crops were highly dependent on the market. So, AGIs will have greater effects on these products.

*4.5. Potential Mechanisms*

4.5.1. Industrial Value-Added

The effects of AGIs on industrial value-added are shown in Table 5. In column (1) of Table 3, we capture the effects of AGIs on value-added in primary industries. This shows that GDP in the agricultural sector increased by 1% when AGIs were added. It shows that AGIs had an obvious value-added effect on economic development and $H_2$ was valid.

However, this effect can be affected by quality or quantity. Therefore, we analyzed the influence of AGIs on the quantity and quality of agricultural products. In columns (2)–(5) of Table 5, we further analyzed the impact of AGIs on the yield of major agricultural productions. This demonstrated that AGIs had a significant positive effect on the yield of grain, cotton, and meat, while the effect on oil was relatively weak. In columns (6)–(7) of Table 5, we capture the effects of AGIs on fertilizer inputs and agricultural machinery usage. AGIs significantly inhibited the use of fertilizer and increased the total power of agricultural mechanization. This implies that AGIs can effectively improve the quality of agricultural products and improve of production efficiency. In the past few years, scholars often assumed that the quality and quantity of products were in competition when analyzing the economic outputs. However, AGIs are an important tool for improving market information asymmetry, and can encourage farmers to ensure the quality of agricultural products and increase agricultural production.

**Table 5.** Effects of AGIs on industrial value-added.

| | (1) | (2) | (3) | (4) | (5) | (6) | (7) |
|---|---|---|---|---|---|---|---|
| | **Value Added in Primary Industry** | **Output** | | | | **Input** | |
| | | **Grain** | **Cotton** | **Oil** | **Meat** | **Fertilizer** | **Power of Agricultural Machinery** |
| AGI | 0.010 *** | 0.013 *** | 0.022 *** | 0.008 ** | 0.013 *** | −0.017 *** | 0.007 *** |
| | (0.001) | (0.002) | (0.007) | (0.004) | (0.002) | (0.005) | (0.002) |
| Control variables | √ | √ | √ | √ | √ | √ | √ |
| Year effects | √ | √ | √ | √ | √ | √ | √ |
| County effects | √ | √ | √ | √ | √ | √ | √ |
| _cons | 10.947 *** | 11.929 *** | 6.248 *** | 6.613 *** | 9.953 *** | 9.320 *** | 1.076 *** |
| | (0.052) | (0.071) | (0.383) | (0.152) | (0.069) | (0.208) | (0.073) |
| N | 33,659 | 28,521 | 13,161 | 32,370 | 30,102 | 18,265 | 31,449 |
| $R^2$ | 0.959 | 0.947 | 0.879 | 0.875 | 0.933 | 0.847 | 0.920 |

Standard errors in parentheses *** $p < 0.01$, ** $p < 0.05$.

### 4.5.2. Factor Agglomeration

Table 6 analyzes the influence of AGIs on agglomeration of three types of factor: population, financial, and industrial. In columns (1)–(3) of Table 6, we investigate the impact of AGIs on the employers in the three industries. The results show that AGIs promoted an increase in employers in agriculture, but not in industry and tourism, and that although AGIs intensified population agglomeration, this occurred within the agricultural part.

**Table 6.** Effects of AGIs on factor agglomeration.

| | (1) | (2) | (3) | (4) | (5) |
|---|---|---|---|---|---|
| | **Population Agglomeration** | | | **Financial Agglomeration** | **Industrial Agglomeration** |
| | **Agriculture** | **Industry** | **Tourism** | **Credits Scale** | **Number of Enterprises** |
| AGI | 0.002 * | −0.006 * | −0.005 ** | 0.005 *** | 0.0048 *** |
| | (0.001) | (0.003) | (0.003) | (0.002) | (0.0004) |
| Control variables | √ | √ | √ | √ | √ |
| Year effects | √ | √ | √ | √ | √ |
| County effects | √ | √ | √ | √ | √ |
| _cons | 13.196 *** | 8.367 *** | 10.012 *** | 3.864 *** | — |
| | (0.103) | (0.350) | (0.269) | (0.075) | — |
| N | 10,122 | 10,163 | 10,175 | 33,250 | 33,360 |
| $R^2$ | 0.984 | 0.940 | 0.937 | 0.924 | 0.865 |

Standard errors in parentheses *** $p < 0.01$, ** $p < 0.05$, * $p < 0.1$; In model (5), we employ the panel Poisson model.

In column (4) of Table 6, we analyze the effects of AGIs on county-level credit loan balance, which indicates that AGIs will stimulate credits expansion. Column (5) of Table 6 shows that AGIs also encouraged new firms to join in. In fact, most agricultural products are perishable. To ensure the quality of agricultural products with AGIs as much as possible, many companies will set up factories and invest in places close to the agricultural products with AGIs. In this way, capital and talent will agglomerate. This is conducive for the integration of various factors and for economy of scale of economy. Therefore, $H_3$ is convincing.

## 5. Conclusions, Discussions, Policy Implications & Limitations

### 5.1. Discussions

A variety of studies have demonstrated the effects of AGIs on market welfare [18], consumer surplus [19], and producer surplus [9]. Most modern countries take AGIs as an essential means of improving product quality [20] and promoting prosperity in the cultural, tourism, and culinary industries [23]. A healthy market is essential for validating these conclusions. In underdeveloped countries, the market is imperfect, making these results confusing. However, this study draws similar conclusions in China, a developing country. Although AGIs only contribute 0.2–0.4% to per capita economic growth, this influence grows over time. Furthermore, AGIs not only improve the quality and quantity of agricultural products, but also seem to activate the agglomeration of market factors such as population, capital, and enterprises. We believe that with the support of the central government, AGIs can become an important force in promoting economic growth in developing countries.

### 5.2. Conclusions

AGIs benefit agricultural product circulation, promote high-quality development in the agricultural sector, and develop and expand the comprehensive functions of agriculture.

In this paper, we investigated the effects of AGIs on county-level economic development and corresponding mechanisms. The results show that AGIs can accelerate China's per capita GDP at a higher rate. Agricultural value-adding, population, capital, and enterprise agglomerating are potential channels for these effects. In the western area, the main grain-producing area, and other-nations-settled areas, the effects of AGIs are more salient. Moreover, AGIs related to aquatic environments, animal husbandry, and planting products are more beneficial for local economic development.

### 5.3. Policy Implications

Based on the conclusions in this paper, three suggestions can be provided for policymakers.

Firstly, keep accelerating AGI identification. Policymakers should understand facts related to AGIs and assess the effectiveness and issues of AGIs. They also should strictly set product standards and technical specifications of AGIs to ensure high-quality development. Further, they could use media to encourage farmers to propagate AGIs and comprehensively investigate and punish the criminal activities of AGIs.

Secondly, encourage the integrated development of industrial chains. Policymakers should promote deep joint development of agriculture, industry, and tourism. They can promote the smooth flow of factor resources such as scientific and technological innovation, modern finance, and human resources. This way, information, technology, knowledge, talent, and other elements can agglomerate effectively and the spillover effect of AGI's brand value can be guaranteed.

Thirdly, improve AGI support policies in certain regions. For the western region, non-grain main producing areas, and national-minority-settling areas, policymakers must strengthen support for fiscal, investment, finance, technology, and other policies on AGIs, as well as encourage financial institutions such as banks and insurance companies to develop AGI-specific financial products and financing models.

### 5.4. Limitations

Although this article indicates a positive impact of AGIs on aggregate GDP, we still need to pay attention to the impact of AGIs on different participants (such as farmers, government, and enterprises). Can farmers receive more returns from AGIs? Who can better benefit from AGIs between upstream and downstream enterprises? For AGIs, does the Government receive more tax revenue or consume more fiscal expenditure? These questions also need to be discussed by followers using more microscopic data.

**Author Contributions:** Conceptualization, Z.Z., Q.Y. and Y.C.; methodology, software, investigation, and writing—original draft preparation, Z.Z., M.Z. and H.Z.; writing—review and editing, Z.Z., Y.C. and Q.Y.; project administration, Y.C.; funding acquisition, Z.Z. and Y.C. All authors have read and agreed to the published version of the manuscript.

**Funding:** This research was funded by the National Natural Science Foundation of China (72273045) and, the National Social Science Foundation of China (20 and ZD117).

**Institutional Review Board Statement:** Not applicable.

**Data Availability Statement:** All the datasets can be found from (https://cnki.nbsti.net/CSYDMirror/trade/Yearbook/Single/N2020070182?z=Z009, accessed on 13 May 2024) and (http://www.anluyun.com, accessed on 13 May 2024).

**Conflicts of Interest:** The authors declare no competing interests.

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
