# Peer review of "Little Brands, Big Profits? Effect of Agricultural Geographical Indicators on County-Level Economic Development in China"

_agriculture, doi:10.3390/agriculture14050767_

Round 1

Reviewer 1 Report

Comments and Suggestions for Authors

Dear authors,

The manuscript is presenting interesting data on correlation of geographical indication with economic development. It is well documented and the method was clearly described, as to be replicated for other regions as well. 

However, before the study being ready for publication several minor changes should be done.

Please go throughout the manuscript again and correct the English language as well as phrase arrangement. Some sentences are difficult to understandd, or need to be transform in subtitles. You can find bellow several examples along with their suggestions:

Line 42 :“ Previous primarily analyzed the impact of AGIs on agricultural products from perspectives such as legal disputes, market welfare, and international trade”è previous studies? Or reports?

Line 46 : Firstly, legal disputes over AGIs.==> subtitle

Line 85: The remainder of this article are arranged as follows=: rephrase

Line 381: “AGI will promotes credits scale”è will promote

Line 390: “We find that AGI can the 390 GDP per capita in the country of China grow at a higher rate”. == verb missing

Figure 2 should be better explained. The measurement unit is missing.

It shows that AGI about aquatic, animal husbandry and planting products promote obviously economic develop ment. è these parameters are situated somewhere between 0 and 0.2. Why is “fishes “not mentioned here as well? If there are different categories, then split somehow the figure.

Best regards, and good luck with your manuscript

Comments on the Quality of English Language

Please see the above recommendations. English quality should be improved.

Author Response

#Reviewer 1
Q1: Line 42 :“ Previous primarily analyzed the impact of AGIs on agricultural products from perspectives such as legal disputes, market welfare, and international trade” è previous studies? Or reports?

R1: Thanks for your suggestions. We have modified the sentences.

“Previous studies mostly analyzed how AGIs affected agricultural products from the standpoints of international trade, market welfare and legal conflicts[4,5,6].”

Q2: Line 46 : Firstly, legal disputes over AGIs.==> subtitle

R2: Thanks for your suggestions. We have modified the sentence.

“Firstly, legal conflicts”

Q3: Line 85: The remainder of this article are arranged as follows=: rephrase

R3: Thanks for your kind reminder. We modified the sentence.

“The rest of this article is arranged as follows.”

Q4: Line 381: “AGIs will promotes credits scale”è will promote

R4: Thanks for your suggestions. We modified the sentence, and you can see in the revised paper.

   In columns (4) of Table 5, we analyze the effects of AGIs on county-level credit loan balance, which indicates that AGIs will stimulate credits expansion.

Q5: Line 390: “We find that AGIs can the 390 GDP per capita in the country of China grow at a higher rate”. == verb missing

R5: You are attentive and thanks. We modified the sentence.

“We find that AGIs can accelerate the China's per capita GDP at a higher rate. Agricultural value-adding, population, capital, and enterprise agglomerating are potential channels for these effects.”Q6: Figure 2 should be better explained. The measurement unit is missing.

It shows that AGIs about aquatic, animal husbandry and planting products promote obviously economic development. è these parameters are situated somewhere between 0 and 0.2. Why is “fishes “not mentioned here as well? If there are different categories, then split somehow the figure.

R6: Thanks for your suggestions. We show the different categories in Figure 2 and explain them in detail.

…….. It shows that main-class AGIs about aquatic, animal husbandry and planting products promote obviously economic development. Moreover, compared with planting AGIs, aquatic products and livestock AGI booms at a higher level. We also analyzed the impact of sub-class AGIs on the county economy. It showed that AGIs about edible fungi, fish, meat, grain, and oil crops had more obvious effects, but AGIs about fruit, tea, medicinal materials, vegetables, and honey not. The demand curve for edible fungi, fish, meat, grain, and oil crops has a strong elasticity, it highly depends on the market. So, AGIs will show greater effect on these productions.

Reviewer 2 Report

Comments and Suggestions for Authors

Thank you for the opportunity of reading and reviewing your manuscript. The paper addresses a topic which is important and relevant for the scope of the journal. 

Specific comments:

- the introduction is well written and creates and it includes the relevant information. However, I suggest adding reference to the aim of the reserach and objective/research question

 .- the literature part is relevant and well written. The main articles are referenced. I appreciate the construction of hypotheses

- the methodology should be explained in more details, there are shortcomings right now

- the model is well constructed, although not sophisticated

- I suggest enhancing the discussion section and also the final section about conclusions and limitations

Good luck!

Author Response

#Reviewer 2

Q1: the introduction is well written and creates and it includes the relevant information. However, I suggest adding reference to the aim of the research and objective/research question.

R1: Thank you for your comments. We have reorganized and modified the introduction

…… Considering the effects of AGIs on economic development, Europe, the United States and Australia have established leading positions in AGIs certification in recent years. However, AGIs certification is relative backwardness in Asian countries. Faced with this situation, the Chinese Ministry of Agriculture prioritized AGIs as a key means to promote regional economic development in 2008. In China, how do AGIs affect aggregated economic development? What are the corresponding mechanisms? Few studies have provided rich knowledge related to these questions, which encourages this article.……

Q2: The methodology should be explained in more details, there are shortcomings right now.

R2: Thanks for suggestions, we explain why we adopt this methodology in details of the revised paper.

In this paper, per capita gross domestic and AGIs vary in different years and counties, so we should adopt a model considering time effects and individual effects. Fixed regional characteristics (Geographical location, planting habits, market distance, etc.) and nationwide policy shocks in certain (Agricultural support policies, agricultural insurance policies, and ecological compensation policies) will interfere with the impact of AGIs on per capita GDP. If nationwide policy shocks and fixed regional characteristics are not important for the effects, the random effects model could be better than the fixed effects model. Otherwise, the fixed effects model will be better. The Hausman value is 468.71 (P-value<0.01), which implies that we should employ a two-way fixed effects model to capture the economic contribution of agricultural geographical indications.

Q3: I suggest enhancing the discussion section and also the final section about conclusions and limitations.

R3: Thanks for your kind reminder. We have revised and reorganized the discussion, conclusions and limitations.

     Firstly, we added the discussions.

A variety of studies demonstrate the effects of AGIs on market welfare [18], consumer surplus [19], and producer surplus [9]. Most modern countries take AGI as an essential means of improving product quality [20] and promoting prosperity in the cultural, tourism, and culinary industries [23]. A healthy market is essential for vali-dating these conclusions. In underdeveloped countries, the market is imperfect, making these results confusing. However, this study draws similar conclusions in China, a developing country. Although AGIs only contributes 0.2-0.4% to per capita economic growth, this influence grows over time. Furthermore, AGIs not only improve the quality and quantity of agricultural products, but also seem to activate the agglomeration of market factors such as population, capital, and enterprises We believe that with the support of the central government, AGIs can become an important force in promoting economic growth in developing countries.

Secondly, we reorganized the conclusions.

AGIs benefit agricultural product circulation, promote high-quality development in the agricultural sector, develop and expand the comprehensive functions of agriculture. In this paper, we investigate the effects of AGIs on county-level economic development and corresponding mechanisms. The results show that AGIs can accelerate the China's per capita GDP at a higher rate. Agricultural value-adding, population, capital, and enterprise agglomerating are potential channels for these effects. In the western area, main grains producing area and other nations settled areas, the effects of AGIs are more salient. Moreover, AGIs related to aquatic, animal husbandry and planting products are more beneficial for local economic development.

Thirdly, we added the limitations.

Although this article indicates a positive impact of AGIs on aggregate GDP, we still need to pay attention to the impact of AGIs on different participants (such as farmers, government and enterprises). Can farmers receive more returns from AGIs? Who can better benefit from AGIs between upstream and downstream enterprises? For AGIs, does the government receive more tax revenue or consume more fiscal expenditure? These questions also need to be discussed by followers using more microscopic data.

Thank you again for your attentions and comments.
